# 6′-Sialyllactose Ameliorates In Vivo and In Vitro Benign Prostatic Hyperplasia by Regulating the E2F1/pRb–AR Pathway

**DOI:** 10.3390/nu11092203

**Published:** 2019-09-12

**Authors:** Bo-Ram Jin, Hyo-Jung Kim, Eun-Yeong Kim, Tae-Wook Chung, Ki-Tae Ha, Hyo-Jin An

**Affiliations:** 1Department of Pharmacology, College of Korean Medicine, Sangji University, 83 Sangjidae-gil, Wonju-si, Gangwon-do 26339, Korea; wlsqh92@gmail.com (B.-R.J.); hyojung_95@naver.com (H.-J.K.); 2Department of Korean Medical Science, School of Korean Medicine, Pusan National University, Yangsan, Gyeongsangnam-do 50612, Korea; eylove0822@hanmail.net; 3Korean Medical Research Center for Healthy Aging, Pusan National University, Yangsan, Gyeongsangnam-do 50612, Korea; twchung@pusan.ac.kr

**Keywords:** benign prostatic hyperplasia (BPH), 6′-Sialyllactose (6SL), pRB, E2F1, androgen receptor (AR), BPH-1, testosterone-induced BPH rat model

## Abstract

Background: 6′-Sialyllactose (6SL) displays a wide range of the bioactive benefits, such as anti-proliferative and anti-angiogenic activities. However, the therapeutic effects of 6SL on benign prostatic hyperplasia (BPH) remain unknown. Methods: Six-week-old male Wistar rats (*n* = 40) were used for in vivo experiments. All rats were castrated and experimental BPH was induced in castrated rats by intramuscular injection of testosterone, with the exception of those in the control group. Rats with BPH were administrated finasteride and 0.5 or 1.0 mg/kg 6SL. Furthermore, the inhibitory effects of 6SL on human epithelial BPH cell line (BPH-1) cells were determined in vitro. Results: Rats with BPH exhibited outstanding BPH manifestations, including prostate enlargement, histological alterations, and increased prostate-specific antigen (PSA) levels. Compared to those in the BPH group, rats in the 6SL group showed fewer pathological changes and normal androgen events, followed by restoration of retinoblastoma protein (pRb) and cell cycle-related proteins. In BPH-1 cells, treatment with 6SL significantly suppressed the effects on the androgen receptor (AR), PSA, and E2F transcription factor 1 (E2F1)-dependent cell cycle protein expression. Conclusions: 6SL demonstrated anti-proliferative effects in a testosterone-induced BPH rat model and on BPH-1 cells by regulating the pRB/E2F1–AR pathway. According to our results, we suggest that 6SL may be considered a potential agent for the treatment of BPH.

## 1. Introduction

Human milk oligosaccharides (HMOs) are major components of human breast milk, consisting of a complex mixture. Up to now, more than 130 different HMOs have been identified [1]. Among these, sialylated HMOs are thought to be particularly important components of HMOs that increase the nutritional value of human milk and show beneficial effects [2,3]. Large amounts of sialylated HMOs are present in colostrum (1–3.3 g/L) and 6′-Sialyllactose (6SL) is the most abundant fraction in sialylated HMOs [4]. A broad range of functions have been attributed to HMOs and recent studies have detailed the inhibitory effects of HMOs and sialylated HMOs on tumorigenesis [5,6]. In addition, our previous study reported that 6SL suggested the inhibitory effects of 6SL on cell proliferation [7]. Although 6SL is thought to show anti-angiogenic and anti-proliferative activities, the therapeutic effects and underlying signaling mechanisms of 6SL on benign prostatic hyperplasia (BPH) remain unknown. 

BPH refers to age-related, noncancerous enlargement of the prostate. BPH is caused by the proliferation of epithelial and stromal cells within the transition zone of the prostate [8]. Patients with BPH develop lower urinary tract symptoms (LUTS), which may ultimately affect the overall quality of life [9]. The etiology of BPH has not yet been fully elucidated, although several factors have been identified. Age-related changes in the levels of testicular hormones are the main factors contributing to the genesis and development of BPH. This hypothesis has been clearly proved by many studies, in which testosterone replacement for patients with hypogonadism led to prostate enlargement [10]. Although the exact mechanism through which testosterone causes BPH is still unclear, the androgen/androgen receptor (AR) signaling and its downstream mechanisms in prostatic cells is thought to be involved [11]. 

Transcriptionally, the E2F transcription factor 1 (E2F1)–retinoblastoma protein (pRb) pathway is involved in cell cycle regulation by controlling the expression of genes that are indispensable to entry into the DNA synthesis phase of the cell cycle, such as cyclin E and cyclin A [12]. pRb is a negative regulator of E2F1 that restrains DNA replication by preventing the G1/S transition in the cell division cycle [13]. Although the exact role of the E2F1–pRb pathway has not yet been elucidated in BPH, a previous study reported that loss of Rb, which is correlated with its phosphorylation, was observed in patients with BPH and early-stage prostatic tumorigenesis, suggesting that the E2F1/pRb signaling pathway is one of the underlying molecular mechanisms of prostatic diseases [14].

Treatment for BPH includes surgical management, transurethral microwave thermotherapy (TUMT), minimally invasive therapies, and medical therapy [15]. Among them, medical therapy remains the first choice for the treatment of BPH. The two principal drugs for this disease are α1-adrenergic receptor blockers, including alfuzosin, and 5α-reductase inhibitors, including dutasteride and finasteride. Among them, 5α-reductase inhibitors work by blocking the conversion of testosterone to dihydrotestosterone (DHT), which then leads to prostate size reduction and improvement of symptoms. However, there is still a risk of side effects, such as fatigue, dizziness, and sexual problems, and a higher risk of developing prostate cancer [16,17]. Consequently, substitutional agents for BPH that are more effective and cause fewer side effects are being considered. 

In this study, we used a testosterone-induced BPH rat model and human BPH cell line (BPH-1) to assess the efficacy of 6SL as a candidate for the treatment of BPH. Here, we demonstrated the effects of 6SL on androgen-relative markers and investigated the molecular mechanisms that eventually lead to cell proliferation in BPH.

## 2. Materials and Methods

### 2.1. Animals

Six-week-old male Wistar rats (*n* = 40) were obtained from Daehan Biolink Co. (Daejeon, Korea). All experimental procedures were carried out according to guidelines for the care and use of laboratory animals established by the National Institutes of Health and approved by the Institutional Animal Care and Use Committee (IACUC) of Sangji University (#2017-21). All rats were fed on NIH-41 open formula diet (Zeigler Bros., Inc., Gardners, PA, USA). 

### 2.2. Induction of BPH and Agent Administration

BPH was induced with testosterone propionate (TP) in male Wistar rats via intramuscular injection, and this epithelial hyperplastic model has been used in numerous studies [18,19]. Briefly, rats were divided into five groups (*n* = 8): Group 1—control animals (Con, castrated rats with vehicle: intramuscular injection of ethanol with corn oil); Group 2—rats with BPH induction (BPH); Group 3—rats with BPH orally administrated 5 mg/kg finasteride (Fina); Group 4,5—rats with BPH intraperitoneally administrated 0.5 or 1.0 mg/kg 6SL (6SL 0.5 and 1.0). Castration was performed by removing the testicles and epididymal fat in all rats. Rats were injected with 10 mg/kg testosterone propionate (TP; Wako Pure Chemicals, Tokyo, Japan) alone or along with finasteride or 6SL every day, except on weekends, for 4 weeks. All animals were sacrificed under anesthesis with Zoletil^®^ 50 (intraperitoneal, 20 mg/kg; Virbac, Carros, France) 28 days after the first testosterone injection. The ventral prostate (VP) and dorsolateral prostate (DLP) were excised, weighed, and stored at −80 °C. 

### 2.3. Serum Level of DHT Analysis

Blood samples were collected from all experimental animals, and serum was separated using a Vacutainer tube (Becton Dickinson, Franklin Lakes, NJ, USA). The serum DHT levels were determined using a commercial enzyme-linked immunosorbent assay (ELISA) kit (CUSABIO; Houston, TX, USA). The assay was performed according to the manufacturer’s instructions.

### 2.4. Histological Analysis and Immunohistochemistry

Hematoxylin and eosin (H&E) for histological analysis and immunohistochemistry (IHC) were performed according to a method described previously [20]. Photomicrographs of stained slides were acquired using an ECLIPSE Ni-U microscope (Nikon, Tokyo, Japan). The thickness of the epithelium in the prostate tissue (TETP) was measured using the Leica Application Suite software (LAS ver. 3.3.0; Leica Microsystems, Inc., Buffalo Grove, IL, USA) in all the glands of each group.

### 2.5. Western Blot Analysis

Primary antibodies against AR (catalog number sc-816), pRb (sc-377528), E2F1 (sc-193), cyclin A (sc-751), cyclin-dependent kinase 2 (Cdk2, sc-748), cyclin D1 (sc-753), proliferating cell nuclear antigen (PCNA, sc-56), and β-actin (sc-81178) were purchased from Santa Cruz Biotechnology, Inc. (Dallas, TX, USA). Antibodies against PSA (PB9259) were obtained from BosterBio Technology (Pleasanton, CA, USA). Protein was lysed and extracted from homogenized rat prostatic tissues and harvested prostatic cells. Western blot analysis was performed according to a method described previously [20]. 

### 2.6. Cell Culture and Sample Treatment

Human BPH-1 cells were obtained from the American Type Culture Collection (Manassas, VA, USA). BPH-1 cells were cultured in RPMI 1640 medium (Gibco, Waltham, MA, USA) containing 20% fetal bovine serum (FBS) and 100 mg/mL penicillin-streptomycin (Hyclone, Hyclone, UT, USA). BPH-1 cells were seeded (5 × 10^5^ cells/well) and incubated for 24 h. The cells were treated with various concentrations of 6SL (12.5, 25, and 50 μM).

### 2.7. MTT Assay

To evaluate cell viability, BPH-1 cells were seeded in 96-well plates (1 × 10^5^ cells/well). After 24 h, cells were treated with various concentrations of 6SL. The following day, MTT solution was added to each well for 2 h, and medium were removed and replaced with dimethyl sulfoxide (DMSO) to measure the formazan concentration. Live cell viability was monitored by measuring absorbance at a wavelength of 540 nm using an Epoch microplate reader (Biotek, Winooski, VT, USA).

### 2.8. Statistical Analyses

The data are expressed as means ± standard deviations (SD) for triplicate experiments. Statistical significance was evaluated via one-way analysis of variance (ANOVA) and multiple-comparison correction based on Tukey’s method, and *p*-values < 0.05 were regarded as statistically significant; *p*-values were calculated using the software GraphPad Prism 5 (GraphPad Software, La Jolla, CA, USA).

## 3. Results

### 3.1. 6SL Attenuated Prostate Enlargement in a TP-Induced BPH Rat Model

After sacrifice, the VP and DLP were dissected and the macroscopic features of BPH were evaluated. Rats with BPH had notably enlarged prostates and showed hyperemia compared to rats in the Con group. However, the Fina, 6SL 0.5, and 6SL 1.0 groups reduced BPH-induced prostate enlargement (Figure 1A, Appendix A). Prostate weight (PW) is commonly measured to predict the development of BPH. As shown in Figure 1B,C, Appendix A, prostate weight and the ratio of PW to body weight (PW/BW index) in rats in the BPH group significantly increased compared to that in the Con group. In contrast to rats in the BPH group, those in the Fina, 6SL 0.5, and 6SL 1.0 groups showed significantly reduced PW and PW/BW index. Abnormal levels of DHT, metabolite of testosterone, may associate with BPH in animal and human studies. Rats in the BPH group had a significantly increased level of DHT compared to rats in the control group. Administration of Fina, 6SL 0.5, and 6SL 1.0 clearly decreased the concentration of DHT in comparison to the BPH group (Figure 1D, Appendix A).

### 3.2. 6SL Ameliorated Prostatic Hyperplasia by Regulating AR Signaling in a TP-Induced BPH Rat Model

To explore the effects of 6SL on histological changes in the prostate gland, H&E staining was performed. As can be seen from Figure 2A, Appendix A rats in the BPH group displayed prostatic hyperplasia signs, including thickened muscle layer, multilayered epithelium, and reduced glandular luminal area in comparison to those in the Con group. However, administration with Fina, 6SL 0.5, and 6SL 1.0 notably attenuated TP-induced hyperplastic patterns. In addition, we measured TETP for objective assessment in histological analysis. The TETP index in BPH group rats was significantly increased (up to 4.77-fold higher) compared to that in rats in the Con group. TETP indexes in rats in the Fina, 6SL 0.5, and 6SL 1.0 groups were significantly lower than those in rats in the BPH group by 58.92, 62.15, and 47.65%, respectively (Figure 2B, Appendix A). AR signaling is correlated with cell proliferation in prostatic epithelial cells, thus promoting BPH development. PSA, the downstream target gene of AR, is a representative biomarker for the progression of prostatic cancer and BPH [11]. Levels of AR and PSA were higher in the BPH group than those in the Con group, whereas Fina, 6SL 0.5, and 6SL 1.0 administration significantly suppressed these levels. Expression of PCNA, which is mediated by androgens at the transcriptional level, was markedly increased in the BPH group compared to that in the Con group. However, Fina, 6SL 0.5, and 6SL 1.0 administration effectively suppressed PCNA protein expression (Figure 2C, Appendix A).

### 3.3. 6SL Suppressed E2F1–pRb Pathway Signaling and Induced Cell Cycle Arrest at the G1 and S Phases in a TP-Induced BPH Rat Model

Increased activity of the transcription factor E2F1 is thought to facilitate the development of benign and malignant disease through cellular proliferation. pRb expression in prostatic tissues taken from each experimental group was examined via IHC. pRb expression was higher in the BPH group than that in the Con group. However, administration of Fina, 6SL 0.5, and 6SL 1.0 decreased pRb expression (Figure 3A, Appendix A). As shown in Figure 3B, Appendix A elevated pRb and E2F1 expression in the BPH group was also observed via Western blot analysis. Fina, 6SL 0.5, and 6SL 1.0 administration notably suppressed pRb and E2F1 protein expression. In addition, the inhibitory effects of 6SL on G1 and S phase cell cycle regulatory proteins were investigated. Elevated cyclin A, Cdk2, and cyclin D1 protein expression stimulated by testosterone treatment was significantly suppressed by Fina, 6SL 0.5, and 6SL 1.0 administration (Figure 3C, Appendix A).

### 3.4. 6SL Abrogated Androgen-Relative Protein Expression in Human BPH Epithelial Cells

BPH-1 cells are derived from an elderly man with BPH and metabolize prostatic androgens. To evaluate the cytotoxicity of 6SL on BPH-1 cells, we performed an MTT assay. As shown in Figure 4A, Appendix A, treatment with 6SL (3.125–50 μM) showed no toxicity in BPH-1 cells. Considering that androgen/AR signaling plays a paramount role in the development and progression of BPH, we investigated whether 6SL inhibited the molecular target of BPH via androgen-dependent signaling. AR and PSA were overexpressed in BPH-1 cells. In contrast, treatment of BPH-1 cells with 6SL (12.5, 25, and 50 μM) for 24 h significantly suppressed the expression of AR and PSA, showing nonspecific cytotoxicity and the specific inhibitory effects of 6SL on BPH-related key markers (Figure 4B, Appendix A).

### 3.5. 6SL Repressed E2F1/pRb Signaling by Regulating G1 and S Checkpoint Protein

To investigate the molecular mechanism by which 6SL inhibited BPH-related protein expression, we investigated E2F1/pRb pathway regulation by 6SL. Previous studies reported that the expression of E2F1-related target genes directly involved in cellular proliferation and transcriptional expression was activated by androgens [21]. As shown in Figure 5A, Appendix A, BPH-1 cells showed higher expression of E2F1 and pRb, and treatment with 12.5, 25, and 50 μM 6SL led to decreased E2F1 and pRb expression. Cyclin A binding with Cdk2 is required for cell cycle S phase progression, and cyclin D is involved in regulating G1 and G1/S phase transitions [22]. 6SL treatments significantly inhibited cyclin A, Cdk2, and cyclin D protein expression compared to that in vehicle-treated cells (Figure 5B, Appendix A).

## 4. Discussion

The androgen–AR axis is crucial in the development and function of the prostate and influences prostatic pathogenesis. Androgens, such as testosterone and DHT, act as ligands and bind to the AR. The androgen/AR complex undergoes dimerization, phosphorylation, and nucleus translocation, followed by regulation of gene expression through various mechanisms. Elevated AR expression has been observed in benign and malignant prostate disease, showing both hyperproliferation and hyperplasia. Emerging evidence has indicated the role of AR in BPH development, and targeting androgen/AR signaling is considered an important therapeutic approach [11]. In the pharmaceutical fields, androgen deprivation therapy (ADT) that reduces androgen levels and/or blocks the action of AR forms the basis of current BPH treatment. Among such agents, 5α-reductase inhibitors have shown successful outcomes in the treatment of BPH. Tempany et al. [23] reported that patients with BPH treated with finasteride showed decreased prostate size, suggesting that suppression of androgen conversion was tightly linked to decreased prostatic volume. A great deal of previous research has ruled out the efficacy of synthetic androgens and these therapies also displayed considerable consequences, including sexual dysfunction, allergic reactions, increased cardiovascular risk, and prostatic disease progression [17]. Currently, as part of the efforts to develop alternatives and minimizes side effects, naturally occurring products such as saw palmetto, β-sitosterol, and cernilton have gained attention [24]. Considering that BPH is a chronic disease, patients desire dietary supplements as a long-term, effective, and safe strategy. However, as scientific evidence regarding the efficacy and safety of supplements is still limited, there is need to demonstrate the pharmacological functions and underlying molecular mechanisms of such agents. 

Sialylated HMOs have potential nutritional functions, such as benefits to the neonate, resistance to pathogens, immune action, and prebiotic effect [25,26,27]. Therefore, sialylated HMOs have received increasing attention for the study of functions and for the production that could mimic the nutritional value of sialylated HMOs. 6SL, a dominant sialylated HMO, has recently become mass-produced and commercially available. Many researchers have focused on anti-proliferation effects of 6SL, as well as its effect on anti-inflammation. Previous studies have shown anti-carcinogenic and anti-angiogenic properties that might further act on inhibition of cell proliferation. 6SL has also exhibited anti-inflammatory and immunosuppressive effects on epithelial cells via inhibition of NF-κB [28]. With the potential involvement of inflammation in the etiology of androgen-induced BPH, many researchers seek to find an anti-BPH treatment. As 6SL has anti-inflammatory and anti-proliferative effects, additional experimental studies of 6SL are warranted to prove its possibility as a potential agent for treatment of BPH. 

In this study, we investigated the effects of 6SL and its action mechanism in a TP-induced BPH rat model and on human BPH-1 cells. Considering that BPH is characterized by hormonal disturbance and pathological proliferation, we established a TP-induced BPH rat model, which has been widely used for BPH research. In our study, rats with TP-induced BPH showed pathological alterations, including swollen and bloodshot prostates, and increased PW indexes and DHT level compared to normal rats. However, administration of 6SL and Fina inhibited prostate enlargement via controlling androgen (Figure 1). Using microscopic analysis, we confirmed the therapeutic effects of 6SL on pathologically-activated epithelial cell proliferation in the prostate tissue of rats with BPH (Figure 2). To understand how 6SL inhibits prostate enlargement, we elucidated the underlying mechanisms of 6SL in BPH. AR, PSA, and PCNA were overexpressed in rats with TP-induced BPH, whereas Fina and 6SL administration suppressed protein expression, implying that the inhibitory effects of 6SL are associated with androgen/AR signaling-dependent hyperproliferation. Overexpression of PCNA, a cell proliferation marker and a key factor in DNA replication during the S phase of the cell cycle, correlates directly with prostate tissue proliferation [29].

Here, IHC revealed that 6SL administration notably reduced the effects of pRb expression compared to that in the TP-induced BPH group (Figure 3A). In accordance with this, 6SL significantly reduced the overexpression of pRb and E2F1, suggesting that the therapeutic effects of 6SL on BPH may be dependent on AR-related E2F1 regulation inhibition (Figure 3B). Previous published reports suggested that AR activity was regulated by various coactivators, including cyclin D1, and that AR-medicated cell cycle activation precipitated the transcription of androgen-sensitive genes [30]. A series of cell cycle activation steps during cell cycle phases, including the association of cyclin D1 with Cdk4 in the middle-to-late G1 phase, followed by the association of cyclin E with Cdk2 in G1/S, allow progression to early S phase and phosphorylate Rb [31]. Meanwhile, cyclin A–Cdk2 complexes are required for entry into the S phase, and are critical factors for cell proliferation [32]. Data from rats with TP-induced BPH showed the upregulation of cyclin A, Cdk2, cyclin D1, and Cdk6 protein expression, whereas administration of Fina and 6SL reduced the expression of cyclin–Cdk complexes (Figure 3C). Inhibition of cell cycle progression by 6SL may be attributed to its anti-proliferative properties, displayed in rats with TP-induced BPH, and by its regulation of the pRB/E2F1–AR network.

In our study, we established an in vitro experiment using the BPH-1 cell line, derived from human BPH epithelial cells. BPH-1 cells were used to investigate the roles of androgens in the pathogenesis of BPH, and to assess the effect of agents on the expression of androgen-sensitive genes. As shown in Figure 4, treatment with 6SL did not affect cell viability in human BPH-1 cells. However, as expected, treatment of BPH-1 cells with 6SL reduced the expression of androgen-related proteins, AR, and PSA. Cell proliferation regulation by androgens is accompanied by major changes in E2F-regulated transcription [33]. Our results also revealed overexpression of pRb/E2F1, cyclin A, Cdk2, and cyclin D1 proteins in BPH-1 cells, whereas 6SL treatment downregulated the expression of cell cycle regulatory proteins, consistent with inhibition of pRb/E2F1 expression. Although 6SL did not inhibit prostatic cell growth, pRB/E2F1–AR protein expression was significantly inhibited, supporting the non-cytotoxic but specific inhibitory effects of 6SL on the pathological or abnormal expression of BPH-related genes.

## 5. Conclusions

In conclusion, 6SL displays protective properties in models of TP-induced BPH in rats and BPH-1 cells by regulating androgen-dependent proliferation responses. These inhibitory effects of 6SL on BPH were associated with decreased expression of AR, PSA, and PCNA. The therapeutic effects of 6SL on BPH both in vitro and in vivo were exerted via suppression of the pRB/E2F1–AR network and regulation of the cell cycle (Figure 6). On the other hand, the suppression of prostate enlargement, achieved using the naturally occurring compound 6SL, is comparable to that achieved with finasteride treatment, which also suggests the possibility of combination to reduce side effects of Fina. To the best of our knowledge, this is the first attempt to understand the pharmacological effects of 6SL on androgen-dependent proliferation in a TP-induced BPH rat model and in BPH-1 cells, suggesting that 6SL was able to alleviate the development of BPH.

## Figures and Tables

**Figure 1 nutrients-11-02203-f001:**
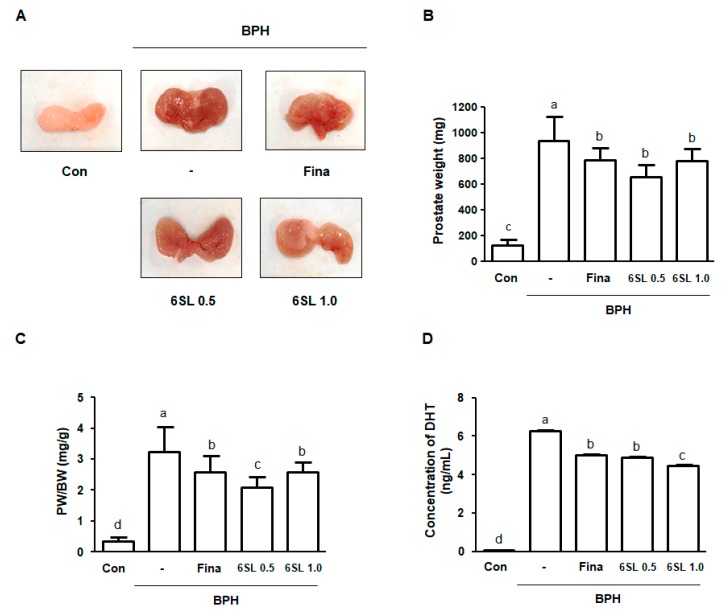
Effect of 6SL on enlarged prostate in a testosterone propionate (TP)-induced benign prostatic hyperplasia (BPH) rat model. (**A**) Representative images showing changes of prostatic tissues from each experimental group are detected. (**B**) Prostate weight (PW) and (**C**) prostate weight to body weight (PW/BW) ratio was measured and analyzed. Prostate weight to body weight (PW/BW) ratio = (Mean value of prostate weight from the experimental group / Mean value of body weight from the experimental group) × 1000. All data are mean ± SD (*n* = 8 per group). (**D**) The concentration of DHT was analyzed using ELISA kit. All data are mean ± SD (*n* = 4). Values with different letters indicate significant differences, *p* < 0.05. Con; control animals, BPH; rats with BPH induction, Fina; rats with BPH orally administrated 5 mg/kg finasteride, 6SL 0.5 and 1.0; rats with BPH intraperitoneally administrated 0.5 or 1.0 mg/kg 6SL.

**Figure 2 nutrients-11-02203-f002:**
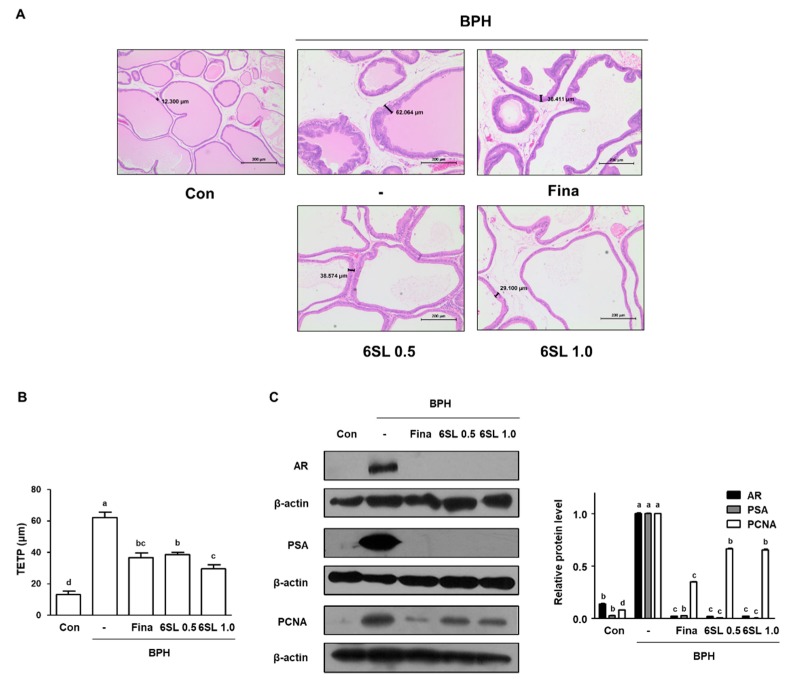
Effect of 6SL on histological change and androgen-relative protein expression in TP-induced BPH rat model. (**A**) Prostatic tissue slides were stained by hematoxylin and eosin (H&E) and observed (magnification × 100). (**B**) Thickness of epithelium tissue from prostate (TETP) was measured and expressed as the mean ± SD of five rats per experimental group. (**C**) The protein expressions of androgen receptor (AR), prostate-specific antigen (PSA), and proliferating cell nuclear antigen (PCNA) in prostatic tissues were determined by immunoblotting. The densities of protein were calculated using ImageJ Software and the relative protein level was normalized to internal control β-actin. Values with different letters indicate significant differences, *p* < 0.05.

**Figure 3 nutrients-11-02203-f003:**
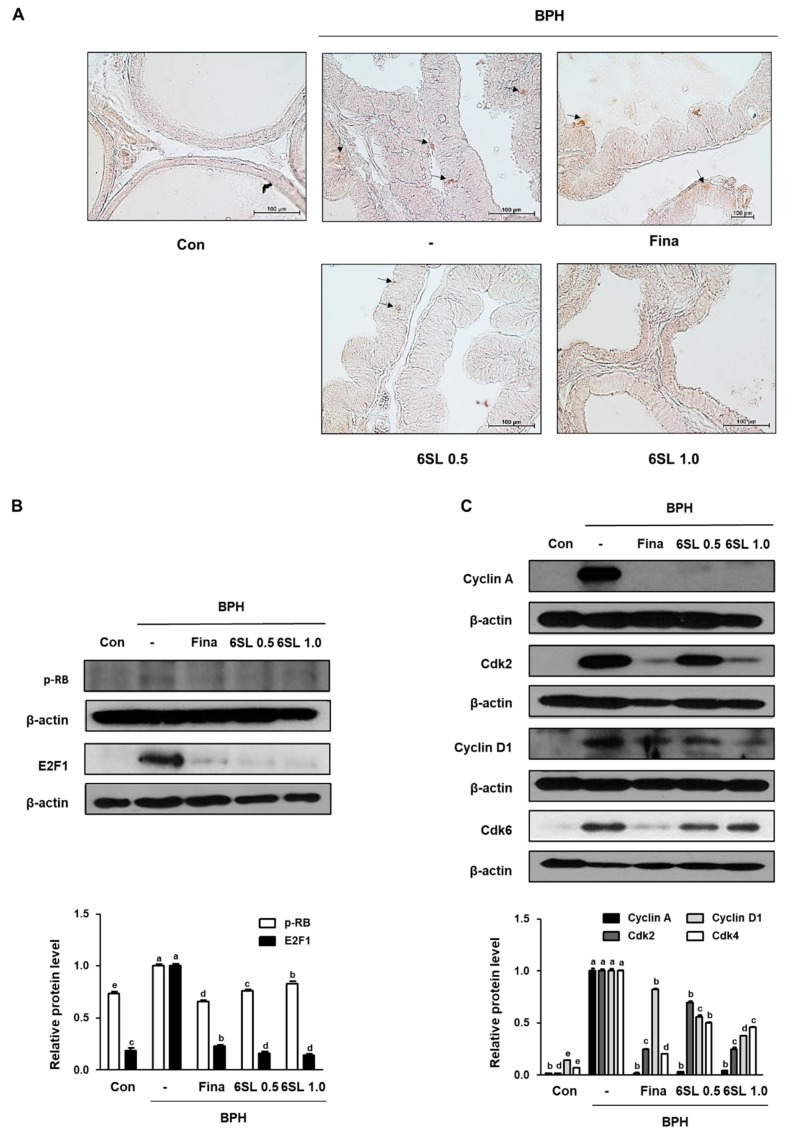
Effect of 6SL on E2F transcription factor 1 (E2F1)-relative and cell cycle protein expression in TP-induced BPH rat model. (**A**) The manifestation of retinoblastoma protein (pRb) in prostatic tissues from rats was shown by immunohistochemistry. The immunoblotting images and quantitative analysis show the protein expression of (**B**) pRb, E2F1, and (**C**) cyclin A, cdk2, cyclin D1, Cdk4. The densities of proteins were calculated using ImageJ Software. Relative protein level represents densitometric values of each protein as ratio to β-actin. Values with different letters indicate significant differences, *p* < 0.05.

**Figure 4 nutrients-11-02203-f004:**
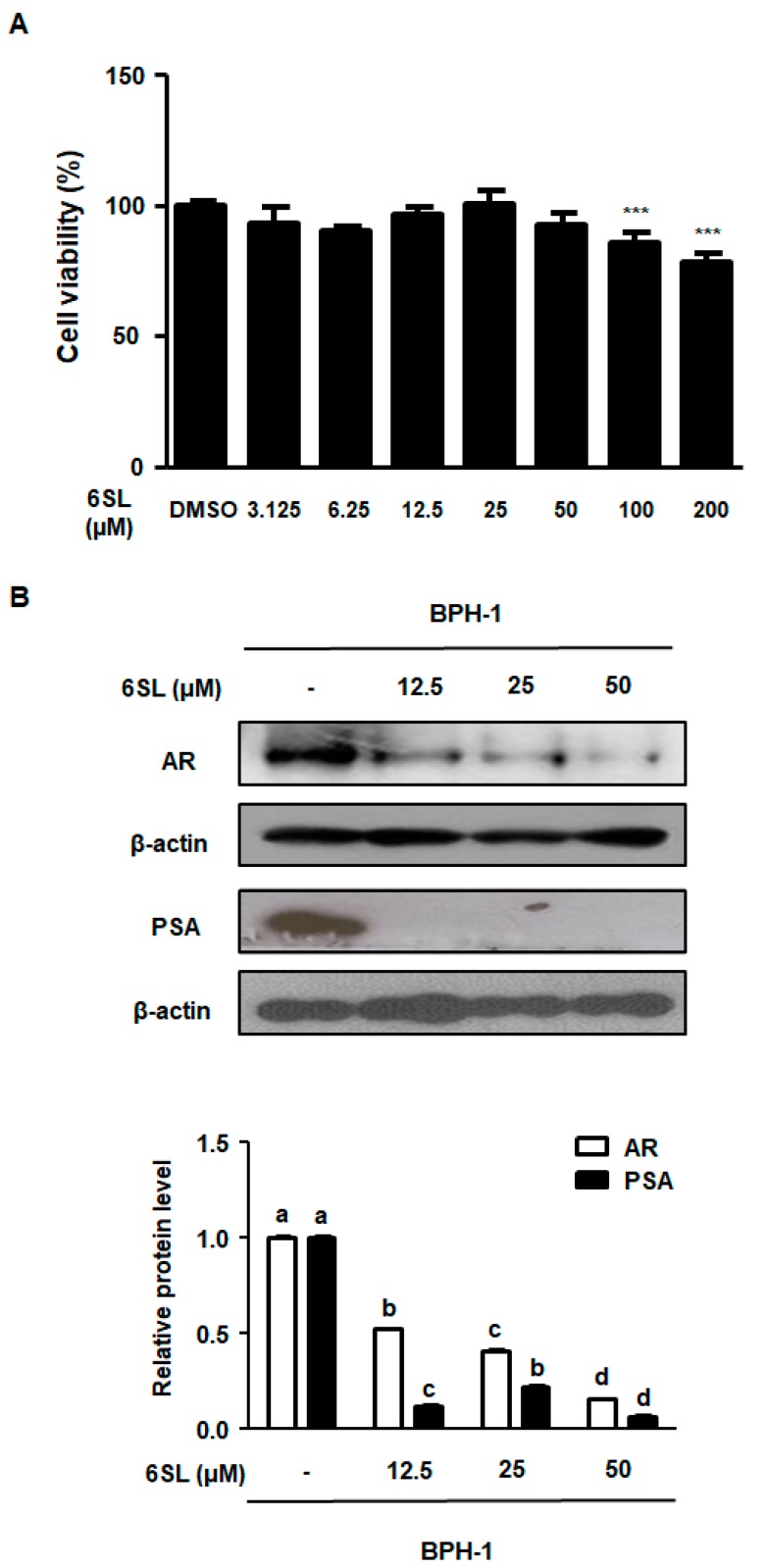
The inhibitory effect of 6SL on BPH cell line (BPH-1) cells’ growth and androgen-relative protein expression. (**A**) BPH-1 cells were treated with various concentration of 6SL for 24 h to assay cell viability. (**B**) AR and PSA protein expression were determined by immunoblotting and relative protein level was analyzed. β-actin was used as an internal control gene. The densities of protein were calculated using ImageJ Software. Values of three separate experiments are represented as mean ± SD. Different letters indicate significant differences, *p* < 0.05. DMSO; Dimethyl sulfoxide.

**Figure 5 nutrients-11-02203-f005:**
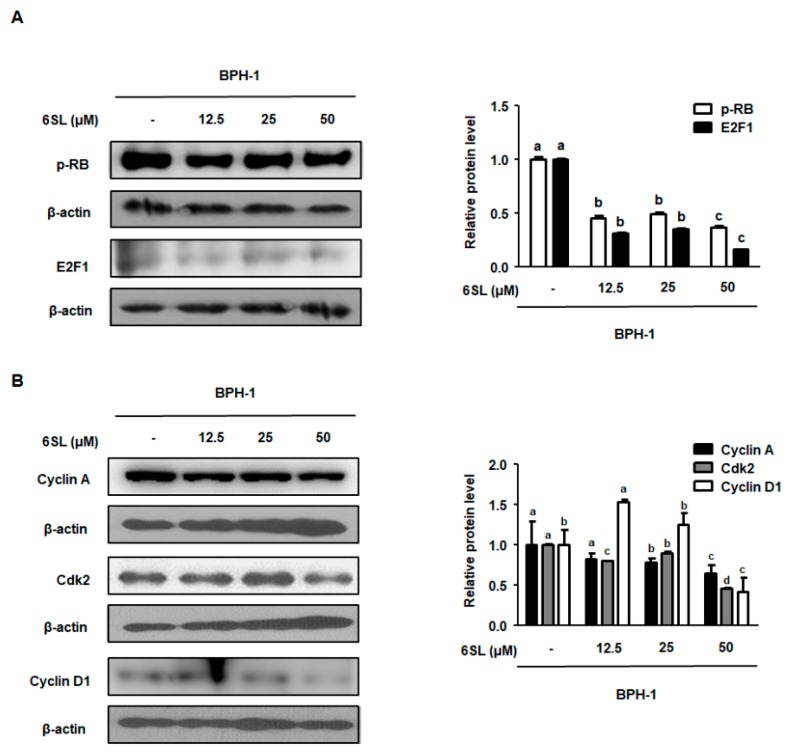
The depressant effect of 6SL on E2F1-dependent cell cycle protein expression in BPH-1 cells. (**A**) pRb, E2F1, and (**B**) Cyclin A, Cdk2, Cyclin D1 protein levels were analyzed by immunoblotting. The densities of proteins were calculated using ImageJ Software. Relative protein level was normalized to β-actin and values of three separate experiments are represented as mean ± SD. Different letters indicate significant differences, *p* < 0.05.

**Figure 6 nutrients-11-02203-f006:**
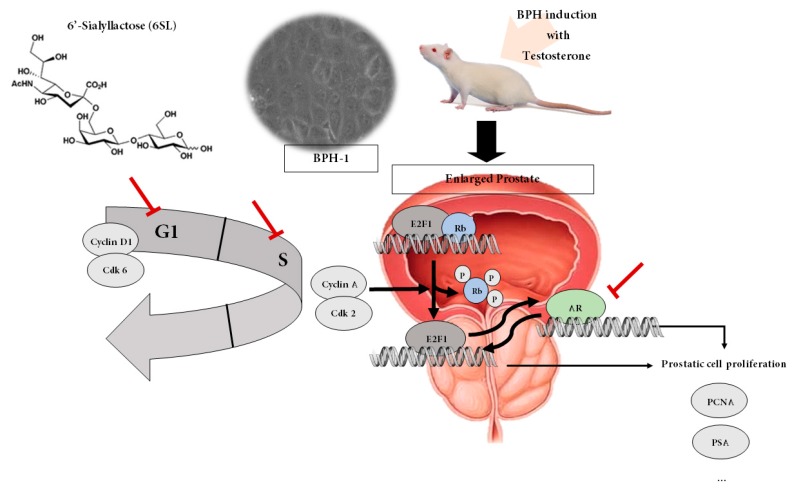
Proposed action mechanism of 6SL in TP-induced BPH rats and BPH-1 cells. 6SL demonstrated anti-proliferative effects in a testosterone-induced BPH rat model and on BPH-1 cells by regulating the pRB/E2F1–AR pathway. 6SL inhibited prostate enlargement via controlling androgen/AR signaling-dependent hyperproliferation. AR influences E2F-regulated transcription that plays a paramount role in cell cycle progression and apoptosis signaling. Here, inhibition of E2F1/pRb and cell cycle progression by 6SL may be attributed to its anti-proliferative properties, supporting its regulation of the pRB/E2F1–AR network.

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
