# Peer review of "6′-Sialyllactose Ameliorates In Vivo and In Vitro Benign Prostatic Hyperplasia by Regulating the E2F1/pRb–AR Pathway"

_nutrients, 2019, doi:10.3390/nu11092203_

Round 1
Reviewer 1 Report
In the present manuscript, authors Jin et al. demonstrated the anti-proliferative effects 6SL in a testosterone-induced BPH rat model and on BPH-1 cells by regulating the 34 pRB/E2F1-AR network. This is a very well written manuscript. Experimental design is sound. Results showed a moderate inhibition of BPH by 6SL but similar to Fina. Per these important studies, 6SL may be considered a potential agent for for the treatment and chemoprevention of BPH. Below are minor points that need to be addressed.
In fig 3A, the IHC image is not clear. PCNA IHC analysis and inclusion of proliferative index will be helpful.
Author Response
_

Reviewer 2 Report
This manuscript describes the effect of 6-Siallactose as a treatment option for prostatic benign hyperplasia. The study includes interesting and useful results, and the analysis of both animals and cell culture gives it extra power. There are still some concerns that need to be addressed:
-Data availability: This journal, Nutrients, state: “Note that publication of your manuscript implies that you must make all materials, data, and protocols associated with the publication available to readers”. It is therefore important that the authors include all weight values and relative protein levels for all the rats and the cell cultures as supplementary. As all experiments were performed on rats or cell culture, there should be not ethical reasons not to include this data.
-Title: Write “pathway” instead of “network”. And should there be a ‘p’ in fron of the Rb, so it’s p-Rb?
-LINE 25: write “.. were used for in vivo experiments”.
-LINE 31: It is unclear what exactly is meant by “abnormal androgen events”
-Introduction: There should be a sentence or two about E2F1 and p-Rb in the introduction. It’s confusing to read the part of the results revolving around these proteins when the reader has no idea why you even looked at those specific proteins.
-LINE 68: From a quick search on finasteride and prostate cancer risk, there seems to be conflicting results. Overall, it appears the drug is decreasing the overall risk of prostate cancer, but the ones that do still get cancer (after using this drug) gets more aggressive cancer. Another (greater) issue is that the reference provided for this statement does not mention prostate cancer at all.
-LINE 82-91: There are several issues that are unclear in this paragraph.
Reference: There should be reference(s) to the “numerous studies” that has used the BHP-model Castration: It is unclear why the rats with induced BHP were also castrated. This needs to be explained. It is also unclear why the control rats were not castrated. Preferably, there should have been an additional control group that also were castrated. The authors also need to shortly describe how castration was performed. Administration: Why did the control group receive an intramuscular injection, while the other groups received an intraperitoneal injection? A proper control group should be treated in the same way, with the exception of the drug. Where group 3 fed finasteride every day? How many rats were there in each group? LINE 82: Change to “BPH was induced with TP in male Wistar rats via injection, ..”. Also, state what type of injection (e.g was it in the tail?).
-Western blot: How was the ‘relative protein levels’ presented in the figures quantified/estimated? This needs to be described.
-Figures: Figures 2A and 3B needs to be a lot bigger, preferably crossing the whole width of the page. It’s impossible to read the annotations and to evaluate the histology. Figure 3B and C needs to have larger font of the letters representing the p-values.
-LINE 129: The word “significantly” should be reserved for the results of statistical tests. Use a different word here, for example “notably” or just remove it all together.
-LINE 130 & 131: I do not believe you can say that anything was “restored” in groups 3-5, based on the overall impression of figure 1, the treated BHP prostates were still more similar to the untreated BHP-rats than to the controls.
LINE 151: Please give examples on histology images of the histology features that are mentioned here.
Thickness of the epithelium tissue from prostate (TETP): The authors need to describe how this TETP indexing was performed. Did you use any image analyzing software? Did you measure all the glands and at different places of each gland?
Figure 2B: I don’t understand the y-axis label. I assume the scale was not in meters (m).
Figure 2C: Should the 6SL 1.0 group really all be labeld with ‘c’? The levels from PCNA looks to be significantly different from AR and PSA.
LINE 174: Write “Increased activity of the transcription factor E2F1 is through to..”
Additional figure: I would ask the authors to include a pathway figure in the discussion that shows the association between the proteins that were measured and discussed. There is a lot of molecular pathways being explained, and it’s very difficult to follow without a figure.
LINE 268-284: I don’t see the point of this paragraph, you need to bring the results of the study more. The next paragraph does this much better and I would remove LINE 268-284 and keep LINE 285-299.
-Manuscript lacks ‘Conflict of interest’-section
Author Response
Dear Editor, Nutrients
We appreciate reviewers’ valuable comments and the opportunity to submit the manuscript, entitled " 6′-Sialyllactose ameliorates in vivo and in vitro benign prostatic hyperplasia by regulating the E2F1/Rb-AR network.” Reviewers’ comments are insightful and we have revised the manuscript accordingly. Below, we address point by point how we respond to reviewers’ comments, and we hope that the revised manuscript is now adequate for publication in Nutrients.
Comments from the Editors and Reviewers:
Reviewer 2
This manuscript describes the effect of 6-Siallactose as a treatment option for prostatic benign hyperplasia. The study includes interesting and useful results, and the analysis of both animals and cell culture gives it extra power. There are still some concerns that need to be addressed:
-Data availability: This journal, Nutrients, state: “Note that publication of your manuscript implies that you must make all materials, data, and protocols associated with the publication available to readers”. It is therefore important that the authors include all weight values and relative protein levels for all the rats and the cell cultures as supplementary. As all experiments were performed on rats or cell culture, there should be not ethical reasons not to include this data.
Response: Thank you for your valuable comment for our study. To follow reviewer’s comment, we included all weight values and relative protein levels for all the rats and the cell cultures as supplementary data.
-Title: Write “pathway” instead of “network”. And should there be a ‘p’ in fron of the Rb, so it’s p-Rb?
Response: To follow reviewer’s comment, we revised title.
-LINE 25: write “.. were used for in vivo experiments”.
Response: To follow reviewer’s comment, we included “.. were used for in vivo experiments”.
-LINE 31: It is unclear what exactly is meant by “abnormal androgen events”
Response: In our study, we mainly confirmed the over-expressions of AR, PSA and PCNA in BPH-induced prostatic tissues as abnormal androgen events. So, in abstract, we mentioned by “abnormal androgen events”. In revised manuscript, we clearly stated by “Compared to those in the BPH group, rats in the 6SL group showed fewer pathological changes and normal androgen events…”.
-Introduction: There should be a sentence or two about E2F1 and p-Rb in the introduction. It’s confusing to read the part of the results revolving around these proteins when the reader has no idea why you even looked at those specific proteins.
Response: To follow reviewer’s comment, we included mention about E2F1 and p-Rb in the introduction (line 61-68).
“Transcriptionally, the E2F transcription factor 1 (E2F1)-retinoblastoma protein (pRb) pathway is involved in cell cycle regulation by controlling the expression of genes that are indispensable to entry into the DNA synthesis phase of the cell cycle, such as cyclin E and cyclin A [12]. pRb is a negative regulator of E2F1 that restrains DNA replication by preventing the G1/S transition in the cell division cycle [13]. Although the exact role of the E2F1-pRb pathway has not yet been elucidated in BPH, a previous study reported that loss of Rb, which is correlated with its phosphorylation, was observed in patients with BPH and early-stage prostatic tumorigenesis, suggesting that the E2F1/pRb signaling pathway is one of the underlying molecular mechanisms of prostatic diseases [14].”
-LINE 68: From a quick search on finasteride and prostate cancer risk, there seems to be conflicting results. Overall, it appears the drug is decreasing the overall risk of prostate cancer, but the ones that do still get cancer (after using this drug) gets more aggressive cancer. Another (greater) issue is that the reference provided for this statement does not mention prostate cancer at all.
Response: Thank you for your pointing out it. Pharmacological targeting of androgen and 5α-reductase has been conducted as a therapeutic strategy for BPH. The activities of androgens and enzyme can be counteracted by adding a 5α-reductase inhibitor such as finasteride. Although this form of androgen blockade confers symptomatic relief, finasteride has been associated with adverse effects, such as sexual dysfunction. In addition, clinical study has added a warning that men undergoing 5α-reductase inhibitor treatment can be exposed to higher risk of high-grade prostate cancer. As part of efforts to substitute synthetic drugs and lessen side effect, in this study, we demonstrated that naturally occurring compound 6′-Sialyllactose is valuable in treating BPH. In addition, to follow reviewer’s comment, we added the reference that mention prostate cancer risk in revised manuscript (line75-56).
“However, there is still a risk of side effects, such as fatigue, dizziness, and sexual problems, and a higher risk of developing prostate cancer [16,17].”
Thompson, I.M.; Goodman, P.J.; Tangen, C.M.; Lucia, M.S.; Miller, G.J.; Ford, L.G.; Lieber, M.M.; Cespedes, R.D.; Atkins, J.N.; Lippman, S.M., et al. The influence of finasteride on the development of prostate cancer. The New England journal of medicine 2003, 349, 215-224.
-LINE 82-91: There are several issues that are unclear in this paragraph.
Reference: There should be reference(s) to the “numerous studies” that has used the BHP-model Castration: It is unclear why the rats with induced BHP were also castrated. This needs to be explained. It is also unclear why the control rats were not castrated. Preferably, there should have been an additional control group that also were castrated. The authors also need to shortly describe how castration was performed. Administration: Why did the control group receive an intramuscular injection, while the other groups received an intraperitoneal injection? A proper control group should be treated in the same way, with the exception of the drug. Where group 3 fed finasteride every day? How many rats were there in each group? LINE 82: Change to “BPH was induced with TP in male Wistar rats via injection, ..”. Also, state what type of injection (e.g was it in the tail?).
Response: Thank you for your critical comment for our study. We added references that used the BHP-model in revised manuscript. To exclude the influence of intrinsic testosterone, castration was performed by removing the testicles and epididymal fat in all rats. In revised manuscript, we shortly describe how castration was performed. In our present study, control rats were castrated. Control group received an intramuscular injection of ethanol with corn oil as vehicle. Testosterone propionate was dissolved in ethanol with corn oil. In abstract of previous manuscript, we described confusedly about induction of BPH. As shown in representative picture of prostate from control group, we can see very small prostate that is only induced by castration. Group 3 rats were administrated with finasteride every day except on weekends for 4 weeks. In our study, six-week-old male Wistar rats (n = 40) were divided into 8 rats per each experimental group. We added mention “(n=8)” in revised manuscript. In addition, to follow reviewer’s comment, we changed the sentence to “BPH was induced with TP in male Wistar rats via injection, ..” and stated what type of injection.
-Western blot: How was the ‘relative protein levels’ presented in the figures quantified/estimated? This needs to be described.
Response: In western blot analysis, we used Image J program and then, estimated density of protein expression. Value of each protein was normalized to value of β-actin and quantified as ‘relative protein levels’. In figure legendary section of revised manuscript, we mentioned a sentence “The densities of protein were calculated using ImageJ Software.” in each figure legends.
-Figures: Figures 2A and 3B needs to be a lot bigger, preferably crossing the whole width of the page. It’s impossible to read the annotations and to evaluate the histology. Figure 3B and C needs to have larger font of the letters representing the p-values.
Response: Thank you for your critical comment for our study. To follow reviewer’s comment, we revised the size of figures and font in revised manuscript.
-LINE 129: The word “significantly” should be reserved for the results of statistical tests. Use a different word here, for example “notably” or just remove it all together.
Response: To follow reviewer’s comment, we clearly deleted word “significantly” and used different word “notably” in revised manuscript.
“Rats with BPH had notably enlarged prostates and showed hyperemia compared to rats in the Con group.”
-LINE 130 & 131: I do not believe you can say that anything was “restored” in groups 3-5, based on the overall impression of figure 1, the treated BHP prostates were still more similar to the untreated BHP-rats than to the controls.
Response: Thank you for your good point. In this study, we focused on preventive and inhibitory effect of 6SL 0.5 and 6SL. To follow reviewer’s comment, we clearly revised sentence.
”However, the Fina, 6SL 0.5, and 6SL 1.0 groups reduced BPH-induced prostate enlargement.”
LINE 151: Please give examples on histology images of the histology features that are mentioned here.
Response:
As shown in this figure, the BPH-induced rats showed histological alterations, such as a thick epithelium, reduced glandular luminal area, and typical hyperplastic patterns compared to that in the Control group. In the same manner, in Fig 2A, we can see the similar tendency in BPH group. In revised manuscript, we have larger figure to look clearly histological change.
Thickness of the epithelium tissue from prostate (TETP): The authors need to describe how this TETP indexing was performed. Did you use any image analyzing software? Did you measure all the glands and at different places of each gland?
Figure 2B: I don’t understand the y-axis label. I assume the scale was not in meters (m).
Response: Thank you for your valuable comment for our study. The authors measured TETP index in all the glands at different places of each group using the Leica Application Suite software (LAS ver. 3.3.0; Leica Microsystems, Inc., Buffalo Grove, IL, USA). And, we displayed representative images from each experimental group. In revised manuscript, we included mention about measurement method of TETP (line109-111).
“The thickness of the epithelium in the prostate tissue (TETP) was measured using the Leica Application Suite software (LAS ver. 3.3.0; Leica Microsystems, Inc., Buffalo Grove, IL, USA) in all the glands at different places of each group.”
In addition, we revised the scale of y-axis label. The sign “μ” was missing.
Figure 2C: Should the 6SL 1.0 group really all be labeled with ‘c’? The levels from PCNA looks to be significantly different from AR and PSA.
Response: In present study, for each marker, statistical significance was evaluated via multiple-comparison correction based on Tukey’s method. Although 6SL 1.0 group was labeled with c on all markers, including PSA, AR and PCNA, this is a sign of significance between different groups, not between markers.
LINE 174: Write “Increased activity of the transcription factor E2F1 is through to..”
Response: To follow reviewer’s comment, we revised correctly.
Additional figure: I would ask the authors to include a pathway figure in the discussion that shows the association between the proteins that were measured and discussed. There is a lot of molecular pathways being explained, and it’s very difficult to follow without a figure.
Response: To follow reviewer’s comment, we included additional figure as Figure 6 in discussion section of revised manuscript.
LINE 268-284: I don’t see the point of this paragraph, you need to bring the results of the study more. The next paragraph does this much better and I would remove LINE 268-284 and keep LINE 285-299.
Response: To follow reviewer’s comment, we deleted some paragraphs in discussion section.
-Manuscript lacks ‘Conflict of interest’-section
Response: To follow reviewer’s comment, we included ‘Conflict of interest’ section.
